# Self-supervised learning improves dMMR/MSI detection from histology slides across multiple cancers

**Charlie Saillard**                                    CHARLIE.SAILLARD@OWKIN.COM
*Owkin, Inc.*

**Olivier Dehaene**                                    OLIVIER.DEHAENE@GMAIL.COM
*Owkin, Inc.*

**Tanguy Marchand**                                    TANGUY.MARCHAND@OWKIN.COM
*Owkin, Inc.*

**Olivier Moindrot**                                    OMOINDROT@GMAIL.COM
*Owkin, Inc.*

**Aurélie Kamoun**                                    AURELIE.KAMOUN@OWKIN.COM
*Owkin, Inc.*

**Benoit Schmauch**                                    BENOIT.SCHMAUCH@OWKIN.COM
*Owkin, Inc.*

**Simon Jegou**                                    SIMON.JEGOU@GMAIL.COM
*Owkin, Inc.*

**Editor:** TBA

## Abstract

Microsatellite instability (MSI) is a tumor phenotype whose diagnosis largely impacts patient care in colorectal cancers (CRC), and is associated with response to immunotherapy in all solid tumors. Deep learning models detecting MSI tumors directly from H&E stained slides have shown promise in improving diagnosis of MSI patients. Prior deep learning models for MSI detection have relied on neural networks pretrained on ImageNet dataset, which does not contain any medical image. In this study, we leverage recent advances in self-supervised learning by training neural networks on histology images from the TCGA dataset using MoCo V2. We show that these networks consistently outperform their counterparts pretrained using ImageNet and obtain state-of-the-art results for MSI detection with AUCs of 0.92 and 0.83 for CRC and gastric tumors, respectively. These models generalize well on an external CRC cohort (0.97 AUC on PAIP) and improve transfer from one organ to another. Finally we show that predictive image regions exhibit meaningful histological patterns, and that the use of MoCo features highlighted more relevant patterns according to an expert pathologist.

**Keywords:** Self-supervised learning, Microsatellite Instability (MSI)

## 1. Introduction

Microsatellite Instability (MSI) is a frequent tumor phenotype characterized by an abnormal repetition of short DNA motifs caused by a deficiency of the DNA mismatch repair system (MMR). MMR deficient tumors (dMMR) result from defects in the major MMR genes, namely *MLH1, MSH2, MSH6, PMS2*. These defects arise either sporadically or as a hereditary condition named Lynch syndrome (LS), predisposing patients to develop cancers in several organs.

Recent studies have shown that immune checkpoint blockade therapy has a promising response in dMMR/MSI cancers regardless of the tissue of origin [Le et al. (2017)]. In 2017, this genomic instability phenotype became the first pan-cancer biomarker approved by the US FDA, allowing the use of pembrolizumab (Keytruda) for patients with dMMR/MSI solid tumors at the metastatic stage [Prasad et al. (2018)].

As of today, systematic MSI screening is only recommended for colorectal cancer (CRC) and endometrial cancer [Svrcek et al. (2019)] where the prevalence is relatively high (10% to 20%), principally to detect LS patients and provide them with adequate follow-up. In early stages of CRC, MSI tumors are associated with good prognosis and resistance to chemotherapy [Sargent et al. (2010)], making the diagnosis of this phenotype all the more essential for patient care and therapeutic decision. dMMR/MSI diagnosis is traditionally done using immunohistochemistry (IHC), polymerase chain reaction (PCR) assays, or next generation sequencing. Those methods can be time-consuming, expensive, and rely on specific expertise which may not be available in every center.

Deep learning based MSI classifiers using H&E stained digital images offer a new alternative for a broader and more efficient screening [Echle et al. (2020)]. In CRC, previous work suggests that the use of such models as pre-screening tools could eventually replace IHC and PCR for a subset of tumors classified as microsatellite stable (MSS) or unstable (MSI) with a high probability [Kacew et al. (2021)]. In other locations where MSI prevalence is lower and screening not done as routine practice, predictive models of MSI status from WSI could be used as efficient pre-screening tools.

In this work, we leverage recent advances in self-supervised learning (SSL) on images. We show that SSL permits to reach state-of-the-art results on colorectal and gastric cancer cohorts from The Cancer Genome Atlas (TCGA), generalizes well on an unseen colorectal cohort (PAIP), and could pave the way for classifiers on locations with low MSI prevalence.

## 2. Related Work

**Expert models**  Several histology patterns on H&E images have been reported to correlate with MSI, such as tumor-infiltrating lymphocytes, lack of dirty necrosis or poor differentiation [(Greenson et al. (2009)]. A series of models based on clinico-pathological features have been developed [Greenson et al. (2009); Jenkins et al. (2007); Hyde et al. (2010); Fujiyoshi et al. (2017); Román et al. (2010)] and reported ROC-AUC performances ranging from 0.85 to 0.92 in various cohorts of patients with CRC. These methods however require time-consuming annotations from expert pathologists and are prone to inter-rater variability.

**Deep learning models**  In a seminal publication, Kather et al. (2019) proved the feasibility to determine the dMMR/MSI status from H&E stained whole slide images (WSI) using deep learning. They trained a first ResNet [He et al. (2016)] to segment tumor regions on WSI, and a second one to predict MSI/MSS status in each tumor tile. Each ResNet was pretrained on ImageNet and the last 10 layers were fine-tuned. Models were trained and validated on different TCGA cohorts and obtained respectively AUCs of 0.77, 0.81 and 0.75 on Colorectal, Gastric and Endometrial formalin-fixed paraffin-embedded (FFPE) datasets.

In a larger scale study focusing on CRC only [Echle et al. (2020)], the same team later trained a model on $n = 6406$ patients, reaching 0.96 AUC (95% CI 0.93–0.98) on an external dataset of $n = 771$ patients. Tumor tissues were manually outlined by pathologists.

Since then, different works based on deep learning methods have been published [Zhang et al. (2018); Cao et al. (2020); Hong et al. (2020); Yamashita et al. (2021); Bilal et al. (2021); Lee et al. (2021)] and are reviewed in Hildebrand et al. (2021). The vast majority rely on networks pretrained on the ImageNet dataset [Deng et al. (2009)] and only the last layers are re-trained or fine-tuned. Most of these papers also rely on tumor segmentation as a first step in their models (either by a pathologist or by a deep learning model).

**Self-Supervised learning**  Over the past few years, rapid progress has been made in the field of SSL using contrastive learning or self-distillation strategies: simCLR [Chen et al. (2020a,b)] , MoCo [He et al. (2020); Chen et al. (2020c, 2021)], BYOL [Grill et al. (2020)], achieving impressive performances on ImageNet without using any labels. Such models have also been shown to outperform supervised models in transfer learning tasks [Chen et al. (2020a); Caron et al. (2021); Li et al. (2021b)].

These advances are of particular interest in medical imaging applications where labeled datasets are hard to collect, and especially in histology where each WSI contains thousands of unlabeled images. There is growing evidence that SSL is a powerful method to obtain relevant features for various prediction tasks from histology images [Dehaene et al. (2020); Lu et al. (2019); Li et al. (2021a); Koohbanani et al. (2021); Gildenblat and Klaiman (2019); Abbet et al. (2021); Srinidhi et al. (2021)]. In this work, we show that self-supervision can be efficiently used to detect dMMR/MSI tumors from histology slides, and outperform ImageNet pretrained models across and between several tumors.

## 3. Methods

### 3.1 Proposed pipeline

First, a U-Net neural network [Ronneberger et al. (2015)] is used to segment tissue on the input WSI and discard the background, as well as artifacts. Second, segmented tissue is divided into $N$ (typically between 10,000 and 30,000) smaller images called tiles. Each tile has a fixed shape of $224 \times 224$ pixels (resolution of 0.5 micron per pixel). Third, the $N$ tiles are embedded into feature vectors of shape $D$ using a pretrained convolutional neural network. Fourth, the $N \times D$ features are aggregated using a multiple instance learning model. This final model is the only one trained using MSI/MSS labels.

In this study, we benchmarked 2 different feature extractors (ResNet-50 pretrained with supervised learning on ImageNet [Deng et al. (2009)] or with SSL on TCGA) and 3 multiple instance learning models (MeanPool, Chowder and DeepMIL).

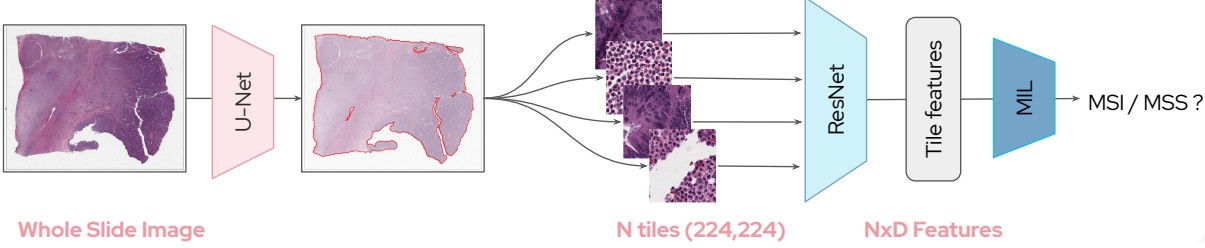

Figure 1: Overview of the proposed pipeline.

**ImageNet feature extraction**  We first extracted features using the last layer of a ResNet-50 pretrained using supervised learning on the ImageNet-1k dataset. We used an auto-encoder to reduce dimension to $D = 256$ because we observed empirically that it significantly improves the performances of Chowder while yielding similar results for MeanPool and DeepMIL. We did not observe similar improvements with MoCo features.

**MoCo feature extraction**  Following Dehaene et al. (2020), we trained several ResNet-50 models using Momentum Contrast v2 (MoCo v2[Chen et al. (2020c)]). We used the exact same parameters and data augmentation scheme but a bigger ResNet backbone was used (the bottleneck number of channels is twice larger in every block). Three different feature extractors were trained: MoCo-CRC using 4.7M tiles from TCGA-CRC, MoCo-Gastric using 4.7M tiles from TCGA-Gastric, and MoCo-CRC-Gastric using the concatenation of the two previous datasets. Each model was trained for 200 epochs (approximately 30 hours) on 16 NVIDIA Tesla V100. The obtained MoCo features have a dimension $D = 2048$.

**MeanPool**  As a baseline multiple instance learning method, we used a simple average pooling of the tile features followed by a logistic regression either with or without $L2$ penalization ($C = 0$, 0.5 or 1).

**Chowder**  We implemented a variant of Chowder [Courtiol et al. (2018)]. A multilayer perceptron (MLP) with 128 hidden neurons and sigmoid activation is applied to each tile's features to output one score. The $R$ ($R = 10$, 25 or 100) top and bottom scores are then concatenated and fed into a MLP with 128 and 64 hidden neurons and sigmoid activations.

**DeepMIL**  We reimplemented the attention based model proposed by Ilse et al. (2018). A linear layer with $N$ neurons ($N = 64$, 128 or 256 here) is applied to the embedding followed by a Gated Attention layer with $N$ hidden neurons. A linear layer followed by a sigmoid activation is then applied to the output.

For both Chowder and DeepMIL, a random subset of $n = 8000$ tiles per WSI was used to accelerate training. All hyperparameters were tuned on the different training sets (see Results). Chowder and DeepMIL were trained with a learning rate of 0.001 using cross-entropy loss and the Adam optimizer [Kingma and Ba (2014)]. When multiple WSI were available for a given patient, we averaged the predictions of the models at test time.

|  | Size | MSI positive | Location | Origin |
|---|---|---|---|---|
| TCGA-CRC | 555 | 78 (14%) | Colon (74%) - Rectum (26%) | US, 36 centers |
| TCGA-CRC-Kather | 360 | 65 (18%) | Colon (74%)- Rectum (26%) | US, 34 centers |
| TCGA-Gastric | 375 | 64 (17%) | Gastric | US, 22 centers |
| TCGA-Gastric-Kather | 284 | 60 (21%) | Gastric | US, 20 centers |
| PAIP | 47 | 12 (26 %) | Colon | Korea, 3 centers |

Table 1: Datasets used in this study.

| Methods | TCGA-CRC-KATHER | TCGA-GASTRIC-KATHER |
|---|---|---|
| *Kather et al. (2019)* | 0.77 (0.62-0.87) | 0.81 (0.69 - 0.90) |
| *Yamashita et al. (2021)* | 0.82 (0.71-0.91) | - |
| *Bilal et al. (2021)* | 0.90 | - |
| Ours - MeanPool | 0.85 (0.76 - 0.94) | 0.78 (0.68 - 0.88) |
| **Ours - Chowder** | **0.92 (0.84 - 0.99)** | **0.83 (0.75 - 0.92)** |
| Ours - DeepMIL | 0.85 (0.75 - 0.94) | 0.79 (0.69 - 0.89) |

Table 2: AUCs on Kather et al. (2019) train/test split. 95% CI are computed following [DeLong et al. (1988)] for our models, and using boostrapping for Kather et al. (2019) and Yamashita et al. (2021).

### 3.2 Datasets

Three different cohorts were used in this study and are summarized in Table 1. For all cohorts, only FFPE images were used.

TCGA-CRC is a dataset of $n = 555$ patients from 36 centers in the US with colorectal tumors. It is a combination of two cohorts from TCGA: TCGA-COAD (colon adenocarcinoma) and TCGA-READ (rectum adenocarcinoma). TCGA-STAD (stomach adenocarcinoma), later referred as TCGA-Gastric, is a dataset of $n = 375$ patients from 22 centers in the US with gastric cancer. For both datasets, MSS/MSI-H labels defined by PCR assays were retrieved using TCGA-biolinks [Colaprico et al. (2016)]. As recommended by ESMO guidelines [Luchini et al. (2019)], MSI-L patients were classified as MSS. TCGA-CRC-Kather and TCGA-Gastric-Kather are variants of respectively TCGA-CRC and TCGA-Gastric datasets published by Kather et al. (2019). They were used here for comparison purposes. These datasets consist of a lower number of cases because MSI-L patients were excluded. The exact same MSI labels were used.

PAIP (cohort from the Pathology AI Platform, http://www.wisepaip.org) is a dataset of $n = 47$ patients from 3 centers in Korea with colorectal tumors. MSS/MSI-H labels were determined using PCR assays.

|  | TCGA-CRC | | TCGA-GASTRIC | |
|---|---|---|---|---|
| Echle et al. (2020) | 0.74 (0.66–0.80) | | - | |
| Kather et al. (2020) | - | | 0.72 | |
| Bilal et al. (2021) | 0.86 | | - | |
|  | ImageNet | MoCo-CRC | ImageNet | MoCo-Gastric |
| Ours - MeanPool | 0.84 (0.05) | 0.87 (0.05) +0.03 | 0.76 (0.04) | 0.82 (0.05) +0.06 |
| Ours - Chowder | 0.81 (0.05) | 0.88 (0.04) +0.07 | 0.73 (0.07) | 0.83 (0.06) +0.11 |
| Ours - DeepMIL | 0.82 (0.05) | 0.88 (0.05) +0.06 | 0.74 (0.01) | 0.85 (0.05) +0.11 |

Table 3: Performances for $3 \times 5$ folds cross-validation (AUC), means, and standard deviations on TCGA-CRC and TCGA-Gastric datasets. Mean, lower and upper bounds on 3 folds are reported by Echle et al. (2020), means on respectively 4 folds and 3 folds are reported by Bilal et al. (2021) and Kather et al. (2020)

## 4. Results

### 4.1 Cross-validations on TCGA-CRC and TCGA-Gastric

We first compared three multiple instance learning models, MeanPool, Chowder and Deep-MIL, using the MoCo-CRC features on the TCGA-CRC-Kather cohort, and the MoCo-Gastric features on the TCGA-Gastric-Kather cohort. For the sake of comparison, we used the exact same train / test split as in [Kather et al. (2019)] and compared our results with the ones published in the literature on this split.

We tuned the hyperparameters of Chowder ($R = 10$, 25 or 100), DeepMIL (size of attention layer = 64, 128 or 256), MeanPool (l2 penalization = 0, 0.5 or 1) and the number of epochs (ranging from 5 to 120) using a grid search on the training sets. For both TCGA-CRC-Kather and TCGA-Gastric-Kather, Chowder model performed best and obtained respectively AUCs of 0.92 and 0.83, achieving state-of-the-art results on these datasets (see Table 2).

To analyze the gain of MoCo features over ImageNet ones, we ran larger cross-validation experiments using 15 distinct splits on the full TCGA cohorts (5 fold cross-validation, repeated 3 times) and reported results in Table 3. For a fair comparison, we tuned the hyperparameters of all models on the training set of the Kather split. In all our experiments, MoCo substantially outperformed ImageNet in both TCGA-CRC and TCGA-Gastric cohorts. Results obtained with MoCo features were also better than the ones previously reported with cross-validation. We also report cross-validation results using center split in Supplementary Table S1.

### 4.2 External validation on CRC dataset PAIP

A limitation of the previous experiments is that ResNet using SSL were pretrained on the full TCGA cohorts, slightly breaking the train/test split independence assumption even if no MSI/MSS labels were used. Thus, we further evaluated MSI detection on an independent cohort of $n = 47$ colon cases from PAIP organisation. We used the median prediction of the

| Methods | IMAGENET | MoCo-CRC |
|---------|----------|----------|
| MeanPool | 0.61 (0.42 - 0.80) | 0.82 (0.66 - 0.99) +0.21 |
| Chowder | 0.86 (0.74 - 0.97) | 0.97 (0.93 - 1.00) +0.11 |
| DeepMIL | 0.83 (0.67 - 1.0) | 0.90 (0.78 - 1.00) +0.07 |

Table 4: AUCs on PAIP for models trained on TCGA-CRC. Bilal et al. (2021) report AUC of 0.98 but no CI interval is given.

| Methods | ImageNet | MoCo-CRC | MoCo-CRC-Gastric |
|---------|----------|----------|------------------|
| MeanPool | 0.76 (0.69-0.82) | 0.67 (0.59-0.74) | 0.76 (0.69-0.82) |
| Chowder | 0.71 (0.64-0.78) | 0.73 (0.66-0.80) | 0.78 (0.73-0.84) |
| DeepMIL | 0.72 (0.66-0.78) | 0.71 (0.64-0.79) | 0.80 (0.75-0.86) |

Table 5: AUCs for models trained on TCGA-CRC and evaluated on TCGA-Gastric with different feature extractors.

ensemble of models trained during cross-validation in the previous experiment. Chowder with MoCo-CRC features yielded an AUC of 0.97 on par with the 0.98 reported in Bilal et al. (2021), and significantly higher than Chowder with ImageNet features (AUC of 0.82, $p = 0.03$ with DeLong's test [DeLong et al. (1988)]). The superiority of MoCo-CRC features was observed for all models (Table 4), confirming that the MoCo-CRC pretrained backbone is a more robust feature extractor than the ImageNet for histology images.

### 4.3 Transfer CRC to Gastric

We assessed the performances for MSI detection when transferring models trained on TCGA-CRC to TCGA-Gastric with three different feature extractors: ImageNet, MoCo-CRC, MoCo-CRC-Gastric. For fair comparison, all hyperparameters were tuned on the training split of TCGA-CRC-Kather (Supplementary Table S2).

MoCo-CRC-Gastric models consistently yielded the best results on TCGA-Gastric with AUCs up to 0.80 (Table 5) while also demonstrating high performances on TCGA-CRC (Supplementary Table S3). Surprisingly, the ImageNet MeanPool model, which performed poorly on PAIP, also obtained a high AUC of 0.76, however significantly lower compared to 0.80 ($p = 0.05$ with DeLong's test). To our knowledge, this is the best performance reported in a transfer setting from one organ to another.

## 5. Interpretability

Within Chowder, each tile is associated with a single score and a MLP is applied to the top and lowest scores of the WSI (Section 3.1). To explore potential differences in interpretability when using ImageNet or MoCo features, a pathologist expert in dMMR tumors reviewed the subset of tiles associated with extreme scores within TCGA-CRC and TCGA-Gastric cohorts, for both feature extractors.

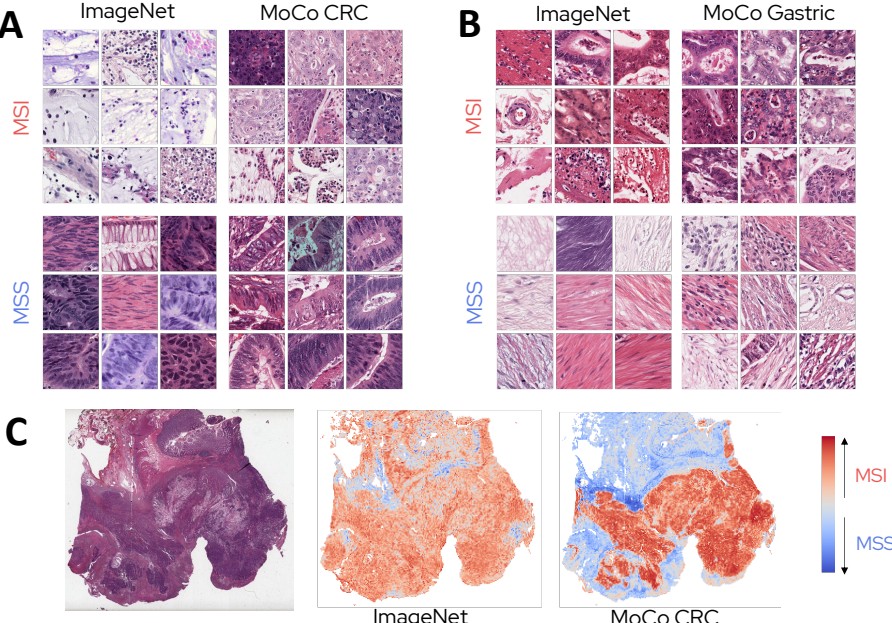

Figure 2: Most predictive regions identified by Chowder A. Tiles most predictive of MSI and MSS phenotypes in CRC. B. Similar for gastric cancer. C. Heatmaps obtained with ImageNet or MoCo-CRC features.

Among the top scored tiles (Figure 2 A, B), the pathologist recognized histological patterns which have been previously described as associated with MSI tumors [Greenson et al. (2009)], including tumor lymphocyte infiltration, mucinous differentiation and presence of dirty necrosis - the latter being more frequently detected in gastric tumors. While ImageNet-based models outputs were based on patterns within and outside tumor regions (such as lymphocyte infiltration in the tumor microenvironment), MoCo-based models clearly focused on tumor epithelium, highlighting both poor differentiation of epithelial cells and tumor infiltrating lymphocytes (TILs) as patterns associated with MSI tumors. Specifically, MoCo models better segmented tumoral regions, as shown by the tile scores heatmap at the WSI level (Figure 2 C).

Overall, lowest scored tiles were unrelated to known MSI patterns, displaying non tumoral tissues such as muscle cells. In the TCGA-CRC cohort, the MoCo-CRC model additionally highlighted the presence of intratumoral epithelial cells without MSI tumor characteristics, showing once again a higher focus on tumor regions, in line with the regions actually inspected by a pathologist when searching for MSI related patterns.

## 6. Discussion

In this work, we showed that feature extractors pretrained using SSL (MoCo V2) on TCGA reach state-of-the-art results for MSI prediction both in colorectal and gastric cancers (Table 2). Extensive cross-validations showed the clear superiority of these models over their

counterparts pretrained using ImageNet (Table 3). In addition, the former generalize better on an external CRC cohort (Table 4). Finally, we observed that using a feature extractor pretrained on several organs using SSL (Table 5) opens the way to both state-of-the-art performances in cross-validation and robust generalization from one organ to another.

There are several limitations to our current study. First, our models could benefit from being trained on more data as shown in the training curve from Echle et al. (2020) (Figure 1.c). Second, our models should be validated on larger cohorts, encompassing different patient populations, treatments (neoadjuvant chemotherapy can impact cell morphology), scanner manufacturer and sample types (resections, biopsies). Third, SSL techniques are evolving rapidly using new architectures such as Vision Transformers [Dosovitskiy et al. (2020)] and more experiments are required to find how to apply them in histology, including taking into account the spatial arrangement of the tiles. However, such experiments require access to extensive computational resources, which limits reproducibility. Finally, in contrast to several concurrent works [Echle et al. (2020); Bilal et al. (2021); Kather et al. (2019)] that fine-tuned the backbones, while our study kept them entirely frozen.

A recent study, Kacew et al. (2021) showed that performances of deep learning models may impact the diagnosis of MSI for patients with CRC. Notably, MSI diagnosis is not routinely done for patients with other solid tumors, missing the identification of candidate patients for immunotherapy. Our results indicate that SSL is a promising solution to develop accurate models for frequent tumor localization such as oesophagus, pancreas, small intestine or even brain, where images are available but MSI prevalence is too low for systematic IHC or molecular testing.

## Acknowledgments

We thank Florence Renaud, pathologist at CHU Lille, for her insightful analysis of the most predictive regions. We thank Patrick Sin-Chan for his corrections of the manuscript.

This work was granted access to the HPC resources of IDRIS under the allocation 2020-AD011011731 made by GENCI.

The results published here are part based upon data generated by the TCGA Research Network: https://www.cancer.gov/tcga.

Regarding the PAIP dataset: De-identified pathology images and annotations used in this research were prepared and provided by the Seoul National University Hospital by a grant of the Korea Health Technology R&D Project through the Korea Health Industry Development Institute (KHIDI), funded by the Ministry of Health & Welfare, Republic of Korea (grant number: HI18C0316).

## Supplementary Tables and Figures

| | | TCGA-CRC | | TCGA-Gastric | |
|---|---|---|---|---|---|
| | Split | ImageNet | MoCo-CRC | ImageNet | MoCo-Gastric |
| MeanPool | CV | 0.84 (0.05) | 0.87 (0.05) +0.03 | 0.76 (0.04) | 0.82 (0.05) +0.06 |
| MeanPool | CV centers | 0.78 (0.10) | 0.85 (0.07) +0.07 | 0.72 (0.12) | 0.85 (0.07) +0.13 |
| DeepMIL | CV | 0.82 (0.05) | 0.88 (0.05) +0.06 | 0.74 (0.01) | 0.85 (0.05) +0.11 |
| DeepMIL | CV centers | 0.79 (0.059) | 0.84 (0.11) +0.05 | 0.73 (0.15) | 0.85 (0.05) +0.12 |
| Chowder | CV | 0.81 (0.05) | 0.88 (0.04) +0.07 | 0.73 (0.07) | 0.84 (0.06) +0.11 |
| Chowder | CV centers | 0.75 (0.15) | 0.83 (0.12) +0.08 | 0.72 (0.08) | 0.86 (0.05) +0.14 |

Table S1: Cross-validation performances (AUC) on TCGA-CRC and TCGA-Gastric. We report mean and standard deviation on $3 \times 5$ folds. We split the data into 5 fold either randomly (CV) or by ensuring that all samples from a center are in the same set (CV centers).

| Feature extractor | Train dataset | MeanPool | Chowder | DeepMIL |
|---|---|---|---|---|
| ImageNet | CRC | C = 1 | R = 100
100 epochs | N = 64
30 epochs |
| ImageNet | Gastric | C = 0.0 | R = 25
30 epochs | N = 64
20 epochs |
| MoCo-CRC | CRC | C = 0.5 | R = 10
10 epochs | N = 32
10 epochs |
| MoCo-Gastric | Gastric | C = 1.0 | R = 100
30 epochs | N = 128
10 epochs |
| MoCo-CRC-Gastric | CRC | C = 0.5 | R = 100
30 epochs | N = 64
10 epochs |

Table S2: Hyperparameters used for the different models. All hyperparameters were tuned on the training sets of the TCGA-CRC-Kather or TCGA-Gastric-Kather cohorts. $C$ refers to L2 penalization coefficient, $R$ to the number of extreme tiles used in Chowder and $N$ to the size of the attention layer in DeepMIL.

| Methods | MoCo-CRC-Gastric |
|---|---|
| MeanPool | 0.86 (0.04) |
| Chowder | 0.88 (0.05) |
| DeepMIL | 0.88 (0.06) |

Table S3: Cross-validation performances (AUC) on TCGA-CRC, for the models trained using the features of MoCo-CRC-Gastric. The result reported are the mean and the standard deviation from 5-fold cross-validation repeated 3 times.

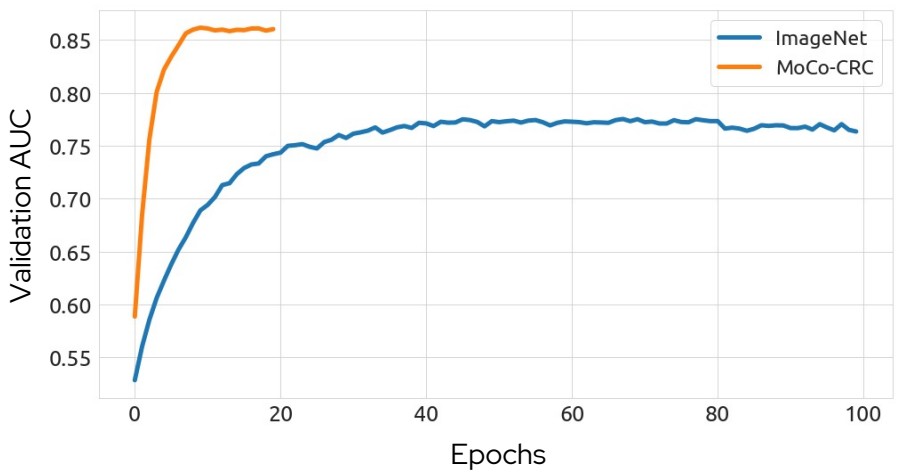

Figure S1: Validation Curves of Chowder with ImageNet and MoCo-CRC features. The plot represents the average validation AUCs per epoch of all runs of the cross-validation for Chowder trained with ImageNet and MoCo-CRC features, depicted in Table 3.

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
