# OpenReview forum: "Self supervised learning improves dMMR/MSI detection from histology slides across multiple cancers"
_MICCAI.org/2021/Workshop/COMPAY — COMPAY 2021_

### Official Review · Reviewer_xfs2 · 2021-08-09
**Self Supervised Learning for MSI Detection**

**Rating:** 5
**Confidence:** 5

**Review:**

The paper's main claim is that self-supervised learning helps improve the performance of MSI detection.

The paper is well written and the problem is introduced well.

Kather et al, 2019 and Bilal et al, 2020 both learn only from tumour patches, whereas you learn from all tissue patches. As far as I'm aware, MSI related features are found only within tumour regions. Please add some comments on this and give reason for why all tissue regions were used.

You have utilised 4.7M tiles from TCGA-CRC and 4.7M tiles from TCGA-Gastric to train the self-supervised approach. One major concern is that there is overlap between the tiles used to train the SSL approach and the TCGA test set. Please can you comment on this. If this is the case, then I think comparing with ImageNet gives an unfair advantage. Especially because the self-supervised was trained on TCGA image patches, I expect that this will make the downstream MIL approach converge a lot faster than when using ImageNet pretrained weights. Could the gain in performance simply be due to the MIL approaches pretrained with ImageNet not yet converging? It would be good to show some training / validation curves to highlight this. The tiles showed in Figure 2 will show some stromal regions showed for ImageNet trained patches, which as I have suggested above should not be diagnostic. This begs the question again - should we just use tumour tiles?

The CNN backbone is frozen and deep features are directly utilised downstream for MIL. It seems obvious that features learned via SSL on TCGA patches will be superior to ImageNet based features for downstream tasks. Instead, it would be good to additionally compare different unsupervised learning approaches on histology data to show whether the features learned via SSL are superior.

"Several histology patterns on H&E images have been reported to correlate with MSI, such as tumor-infiltrating lymphocytes, lack of dirty necrosis or poor differentiation" - please provide references for the above.

Overall, the paper has potential, but the points must be considered above and the experiments conducted to support the claims of the author.

---

### Official Review · Reviewer_gZMG · 2021-08-19
**A momentum contrast based method to classify dMMR/MSI using H&E stained CRC and Gastric tissue sections**

**Rating:** 9
**Confidence:** 4

**Review:**

The authors provide a well written, clinically relevant application example for self supervised learning using H&E stained tissue sections. The excellent AUC observed in the independent external validation data set builds further trust in the proposed method. In addition to the technical component of the work, it would be interesting to learn which are the minimum requirements on specificity/sensitivity for such an H&E based approach compared to the established IHC or PCR methods.

---

### Official Review · Reviewer_TJeN · 2021-08-23
**Solid paper showing improved performance by self-supervision - with one possible poor choice.**

**Rating:** 7
**Confidence:** 4

**Review:**

This paper describes the application of self-supervised learning (SSL) to pathology data to generate more meaningful feature extractors, as quantified by its performance on a multiple-instance learning (MIL) classification task. The authors compare the performance of several MIL approach based on features extracted with (out-of-domain pretrained) ImageNet networks versus (in-domain pretrained) MoCo v2 SSL networks. Their results allow them to conclude that self-supervision improves performance on dMMR/MSI detection in H&E stained whole slide images. The study and paper is similar to recent work by Dehaene et al. 2020 (https://arxiv.org/abs/2012.03583) and seems to confirm their results.

The manuscript is well-written and the study seems carefully executed with relevant cross-validations, validation on an external dataset and suitable statistical analysis.

My only worry with this paper is the fact that the authors have used dimensionality reduction (auto-encoder) on the ImageNet pretrained ResNet50 model to reduce the dimensionality to D=256 (from presumably D=2048), in order "to overcome stability issues". However, such dimensionality reduction has not been applied to the MoCo feature extractor, even though it has the same output dimensionality D=2048. This large (8-fold) difference in dimensionality (as well as the processing by the autoencoder) causes the two feature extraction approaches (ImageNet and MoCo-SSL features) to become less comparable, unnecessarily. Moreover, I am slightly worried that this dimensionality reduction may explain the suboptimal performance of the ImageNet models. The manuscript could be improved by better argumentation for this choice, or mentioning this in the limitations.

---

### Decision · Program_Chairs · 2021-08-25

Accept